# Determining the perceptions and practices of oncologists regarding venous thromboembolism risk assessment in ambulatory cancer patients: A qualitative study

**Marwa Akram Tariq**[1], **Ehab Mudher Mikhael** [2] *

1 Department of Clinical Pharmacy, College of Pharmacy, University of Al-Qadisiyah, Al-Diwaniyah, Iraq,
2 Department of Clinical Pharmacy, College of Pharmacy, University of Baghdad, Baghdad, Iraq

☯ These authors contributed equally to this work.
* ihab.maddr@copharm.uobaghdad.edu.iq

**Data Availability Statement:** All relevant data are within the manuscript and its Supporting Information files (the interview guide is uploaded as supporting file S1 Appendix).

## Abstract

Cancer-associated thrombosis (CAT) can increase morbidity and mortality for cancer patients. Therefore, guidelines recommend predicting VTE risk and thromboprophylaxis for high-risk patients. Many studies critique oncologists' adherence to thromboprophylaxis guidelines for cancer patients. Meanwhile, most of these studies did not discuss in detail the reasons and facilitators for oncologists' adherence to thromboprophylaxis guidelines. Therefore, the current study aimed to explore in depth the perceptions and practices of oncologists working in oncology centers in Baghdad, Iraq, regarding VTE and its risk assessment among ambulatory cancer patients. A qualitative study with face-to-face individual-based interviews was conducted with oncologists working in four major oncology centers in Baghdad, Iraq using a semi-structured interview guide. The guide was developed based on previous relevant literature and validated by a panel of experts. The interviews were conducted from November 2023 to January 2024. Thematic analysis approach was used for data analysis. Thirty-one oncologists were interviewed in this study. Twenty-two of the interviewed oncologists reported that they detect VTE among their cancer patients. 64% of participating oncologists reported that they did not conduct VTE risk assessments for their cancer patients. Only four oncologists reported assessing VTE risk using the Khorana score. 58% of oncologists reported that they prescribe thromboprophylaxis for high-risk patients; meanwhile, only 11% of them reported prescribing anticoagulants in a dose similar to that reported by thromboprophylaxis guidelines. 77% of participating oncologists reported that pharmacists have a significant role in preventing cancer-related thrombosis by helping physicians prescribe a safe and effective prophylactic anticoagulant and in calculating VTE risk scores. In conclusion, CAT is commonly diagnosed among Iraqi cancer patients. VTE risk assessment for ambulatory cancer patients is rarely conducted by oncologists working at Oncology centers in Baghdad, Iraq. The prophylactic anticoagulants were rarely prescribed in appropriate dose and/or duration for patients at high risk of VTE. Pharmacists can help

**Funding:** The author(s) received no specific funding for this work.

**Competing interests:** The authors have declared that no competing interests exist.

oncologists follow thromboprophylaxis guidelines by calculating VTE risk score and recommending a safe and effective dose of appropriate prophylactic anticoagulant.Educating and training oncologists about VTE risk assessment is recommended to enhance their practice in thromboprophlaxis.

## Introduction

Venous thromboembolism (VTE) is a serious, potentially life-threatening condition and a major cause of mortality and morbidity [1,2]. Approximately 20% of all newly diagnosed cases of VTE are cancer patients [3], and an estimated 8% of cancer patients develop VTE within one year after diagnosis or progression of their malignancy [4,5]. Therefore, current guidelines recommend primary prophylaxis for VTE prevention in high-risk patients after assessing the competing risk of bleeding [6,7]. The prediction of VTE in cancer patients often involves the use of risk assessment models or scores (e.g., Khorana Score) based on the presence of certain clinical and laboratory factors [8].

Despite available evidence and existing guideline recommendations for VTE risk assessment of ambulatory cancer patients with primary thromboprophylaxis for identified high-risk patients, several studies found limited utilization of both risk assessment and prescription of primary prophylaxis in clinical practice [9–11]. However, several questions remain regarding assessing risk and prescribing prophylaxis. First, understanding clinician knowledge and understanding regarding guideline recommendations. Second, the frequency of VTE risk assessment and prescription of primary thromboprophylaxis for identified high-risk patients [10]. Therefore, the current study aimed to explore the practices and perceptions of oncologists working in Baghdad, Iraq, regarding cancer-associated VTE, its risk assessment, and prevention among ambulatory cancer patients.

## Methods

### Study design

A qualitative study design was chosen to achieve the study's aim. The study was accomplished through face-to-face individual-based interviews with oncologists. The interviews were guided by semi-structured open-ended questions (S1 Appendix). Probes were used to clarify vague answers and to elicit further comments when necessary. The interview guide was developed by study authors after reviewing relevant literature [9,12]. The guide was reviewed by a panel of experts (Three academic clinical pharmacists with experience in qualitative research and one oncologist). The validated interview guide consisted of eight questions, five (question 1 to 5) designed to assess oncologists practice in detecting and preventing VTE. All obtained answers regarding the type, dose, and duration of prophylactic anticoagulant were evaluated for appropriateness according to the recommendations in thromboprophylaxis guidelines [6,7]. Two questions were developed to assess oncologists' perceptions of the advantages of thromboprophylaxis and pharmacists' role in thromboprophylaxis. The last question was designed to capture any additional comments from participants.

The current study was ethically approved by the ethical committee at the College of Pharmacy/University of Baghdad (approval number of RECAUBCP10620238). The ethical committee accepted waiving written informed consent due to cultural issues and potential fear

from signing documents by most Iraqi individuals. Thus, it was requested that the authors obtain verbal informed consent from study participants.

## Setting and participant recruitment method

To obtain a wide range of oncologists' perceptions and practice toward VTE risk assessment, those working in the outpatient clinics in four of the largest oncology centers in Baghdad, Iraq (Oncology Teaching Hospital, Al-Amal National Hospital, Al-Kadhmiya Teaching Hospital, and Al-Yarmouk Teaching Hospital) were invited by the principal study author to participate in this study [13]. Only physicians with clinical degrees in oncology (high diploma or clinical board) were considered eligible to participate in this study. All eligible oncologists who were interested in participating in the study and provided their verbal informed consent were interviewed [14,15] (Fig 1). To ensure sufficient time for the interview, all study participants were asked to specify a date and time for the interview that could fit with their work schedule. To ensure privacy and confidentiality, all participants were interviewed in a quiet area either in their office or in the consultation area while keeping a distance of at least 5 meters from other physicians or patients.

## Data collection and analysis

The interviews were audio-recorded using a mobile (Xiaomi mi note 10 pro) recorder. Each interview took approximately 10–20 minutes. To achieve the study aims by collecting the desired sample, the interviews were continued from November 2023 to January 2024.

All interviews were coded manually and then used for qualitative data sorting. The coding procedure was established by the first study author (MSc candidate in clinical pharmacy) and reviewed by the second study author (PhD in clinical pharmacy). Any discrepancy in coding was solved after negotiation and reaching consensus. The coding procedure was started by a thorough reading of each interview transcript. A codebook was developed to ensure consistent coding across interviews.

Thematic analysis was conducted based on Braun and Clarke's six steps: "getting to know the comments, generating codes, searching for themes, assessing themes, defining and labeling themes, and finally writing the results" [16].

## Results

Thirty-one oncologists were interviewed in this qualitative study. Most participants (N = 16) were males. The majority (N = 22) had an Iraqi board in clinical oncology. Most participants

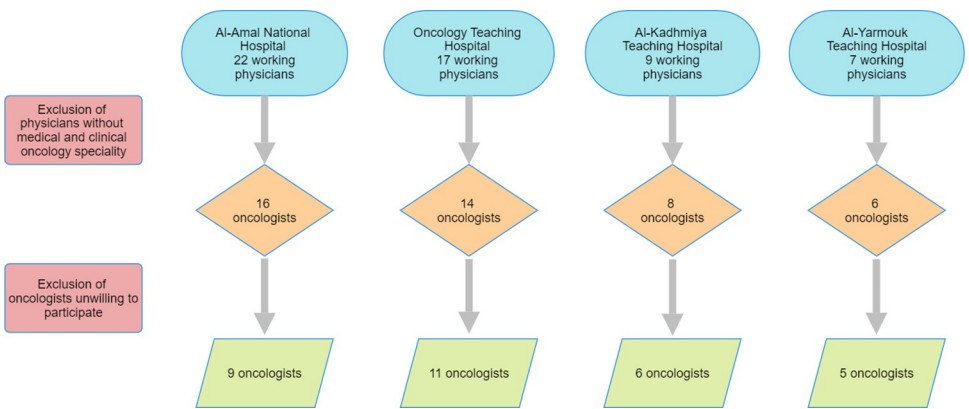

**Fig 1. Flow chart of enrolled study participants.**

**Table 1. Demographics of study participants.**

| Parameter | | Value(mean±SD) |
|---|---|---|
| Age (in years) mean±SD | | 40.74±7.69 |
| Gender | Male n (%) | 16 (51.6%) |
| | Female (%) | 15 (48.4%) |
| Working experience | Less than ten years n (%) | 11 (35.5%) |
| | 10–20 years n (%) | 17 (54.8%) |
| | More than 20 years n (%) | 3 (9.7%) |
| Academic degree | High diploma n (%) | 7 (22.6%) |
| | Iraqi Board n (%) | 22 (71%) |
| | Egyptian Board n (%) | 2 (6.4%) |
| Working place | Al-Amal National Hospital n (%) | 9 (29%) |
| | Oncology Teaching Hospital n (%) | 11 (35.5%) |
| | Al-Yarmouk Teaching Hospital n (%) | 5 (16.1%) |
| | Al-Kadhmiya Teaching Hospital n (%) | 6 (19.4%) |

(N = 20) had at least ten years of clinical experience (Table 1). Further details of study participants are provided in (S1 Table). The generated study themes and subthemes are shown in Table 2.

## VTE among cancer patients

Nearly half of the interviewed oncologists (n = 14) reported that they sometimes detect VTE among their cancer patients. Meanwhile, nine oncologists mentioned that their cancer patients rarely develop VTE, while the last eight participants reported that VTE is commonly developed in their cancer patients. In addition, the study participants reported that patients with certain types of cancer have a high risk of developing VTE (Table 3) such as gastric (n = 17), pancreatic (n = 14), breast (n = 7), lung (n = 7), colorectal (n = 6), gynecological (n = 4), pelvic (n = 4), prostate (n = 2), liver (n = 2), kidney (n = 1), bladder (n = 1), lymphoma (n = 1), and

**Table 2. Study themes.**

| Theme | Subtheme |
|---|---|
| VTE among cancer patients | Prevalence of VTE among cancer patients |
| | Types of cancer that increase the risk of VTE |
| VTE risk assessment | Conduction of VTE risk assessment |
| | Method of VTE risk assessment |
| | Challenges to conducting VTE risk assessment |
| Dealing with cancer patients at high risk for VTE | Prescribing of anticoagulant therapy |
| | Advice about non-pharmacological therapies |
| | Observation of the patient |
| | Referral of the patient to internists and cardiologists |
| Advantages of VTE risk assessment | Patient-related advantages |
| | Physician-related advantages |
| | Institution-related advantages |
| Pharmacists' role in prevention of VTE among cancer patients | Calculating the VTE risk |
| | Recommending an effective and safe thromboprophylaxis therapy |
| | Dispensing therapy |

**Table 3. Types of cancer that reported to be associated with high risk for developing venous thromboembolism according to participated oncologists.**

| Type of cancer | No. of participants (%) |
|---|---|
| Gastric cancer | 17 (54.8) |
| Pancreatic cancer | 14 (45.2) |
| Breast cancer | 7 (22.6) |
| Lung cancer | 7 (22.6) |
| Colorectal cancer | 6 (19.4) |
| Gynecological cancer | 4 (12.9) |
| Pevlic cancer | 4 (12.9) |
| Prostate cancer | 2 (6.5) |
| Liver cancer | 2 (6.5) |
| Renal cancer | 1 (3.2) |
| Bladder cancer | 1 (3.2) |
| Lymphoma | 1 (3.2) |
| Brain cancer | 1 (3.2) |

*Some physician reported more than one type of cancer; therefore summation results in percentages more than 100%.

brain cancer (n = 1). It is important to note that these findings reflect the observations of the participating oncologists among their patients in daily practice, rather than the actual incidence of VTE in these populations. Meanwhile, two oncologists reported that they sometimes detect VTE among patients without regard to the type of cancer; one of them considered metastasis, and the other considered immobility as the most critical risk for developing VTE among cancer patients (n = 1).

> "*VTE is a common condition. About four in ten cancer patients will develop VTE. I often detect VTE among my cancer patients. Malignancy itself is considered a risk factor for developing VTE. VTE mostly occurs in pancreatic cancer followed by gastric cancer*" (P10)

> "*VTE is rare, affecting less than 2% of cancer patients. I diagnosed only one VTE case in the last 49 cancer patients that I have seen. Pelvic cancer is the most common site of cancer that increases the risk for developing VTE*" (P12)

> "*Sometimes I detect VTE; it's not a common condition, and I detect it in about less than 5% of my cancer patients. It's not related to cancer type; it's related to the condition of patients, e.g., wheelchair patients regardless of cancer type*" (P2)

## Risk assessment among cancer patients

The majority of oncologists (n = 20) (65%) emphasized that they did not conduct VTE risk assessments for their cancer patients. Meanwhile, most of these participants refer VTE cases to other specialist physicians when they are developed. On the other hand, eight oncologists emphasized that they conduct a risk assessment only for patients suspected to have a high risk for VTE. The last three oncologists (9.7%) emphasized that they conduct risk assessments for all new cancer cases.

Of these 11 oncologists conducting risk assessment, three mentioned that they depend on clinical examination for risk assessment. Furthermore, eight oncologists reported using the

risk prediction score; four reported using the Khorana score, and the other four used the Wells score. More than half of the oncologists (n = 5) who used risk prediction scores reported that they use clinical website calculators to help them remember score parameters.

> "*I did not assess my cancer patients for VTE risk because there is no such thing in the guideline. Patients with signs and symptoms of VTE are only assessed by Doppler*" (P18)

> "*I always perform a risk assessment for all new cancer patients. I assessed them using the Khorana score because it is an NCCN guideline-validated score. It depends on parameters like the cancer site, complete blood count, and clinical examination. It's easy to apply, and no challenges are associated. I calculate risk for patients by using Medscape*" (P21)

> "*To be honest, there is a score, I do not remember its name, that can be used for VTE risk assessment; however, I do not use it in daily practice. I assess VTE risk among my patients who are newly diagnosed with cancer by clinical examination. In addition, I prefer starting thromboprophylaxis for patients with pancreatic or stomach cancer because such cancer types are risky for VTE development*" (P30)

> "*I often conduct VTE risk assessment for cancer patients who are diagnosed at advanced or metastatic stage (stage 3 or 4). VTE risk assessment can be done by calculating the Wells score. I chose this score because it's practical, easy, and can be performed anywhere using simple parameters. I usually use the medical application to calculate it*" (P25)

## Problems associated with risk assessment

Nearly half of the participating oncologists (n = 15) reported that risk assessment is an easy process and that no problems are associated with it. On the other hand, 16 oncologists reported some obstacles to the process of VTE risk assessment among cancer patients; these include unavailability of medical resources in hospitals (n = 7), limited oncologists' competence (n = 5), and being a time-consuming process that is difficult to perform due to oncologist workload (n = 4).

> "*It is an easy process; I did not face any problems in VTE risk assessment. Also, the diagnosis of VTE is an easy process*" (P11)

> "*Risk assessment is a difficult process that needs special investigations and a specialist physician. So, when I suspect a case of VTE, I refer it to the cardiologist*" (P4)

> "*It is difficult to perform VTE risk assessment for all cancer patients because the process needs time, and I do not have sufficient time to do so due to work overload*" (P16)

> "*Investigations such as Doppler and CT angiography are not always available in hospitals, so the VTE risk assessment is difficult*" (P20)

## Dealing with cancer patients at high risk for VTE

Nine of the interviewed oncologists reported referring high-risk patients to cardiologists. Four participants reported that they did not take any action for high-risk patients and only managed confirmed cases of VTE. Conversely, the majority of oncologists (n = 18) (58%) reported that they prescribe thromboprophylaxis for high-risk patients.

The most commonly reported thromboprophylaxis agent prescribed by those oncologists includes Rivaroxaban (n = 9), Enoxaparin (n = 3), and Aspirin (n = 2). However, four participants reported that they sometimes prescribe Enoxaparin as a thromboprophylaxis agent and

**Table 4. The most commonly prescribed anticoagulants for thromboprophylaxis among ambulatory cancer patients.**

| The most commonly prescribed anticoagulant | No. of oncologists prescribing the anticoagulant | Prescribed in a correct prophylactic dose n (%)^ |
|---|---|---|
| Rivaroxaban | 9 | 1 (11.1) |
| Enoxaparin | 3 | 0 (0) |
| Aspirin | 2 | - # |
| Enoxaparin or Rivaroxaban | 4 | 1 (25)* |

^ Percentage was calculated based on the number of oncologists who prescribe the anticoagulant (not the whole study sample); # Dose not checked since Aspirin is not indicted for thromboprophylaxis

* Accuracy of reported dose was checked for both items.

prescribe Rivaroxaban at other times (Table 4). Of oncologists that prescribe primary thromboprophylaxis, most oncologists (n = 10) prescribe inappropriate doses compared to those recommended by guidelines or prior randomized controlled trials [17,18], while two oncologists prescribed them in appropriate prophylactic doses. Four oncologists reported that they decided on the dose of prophylactic anticoagulant according to patient status, while the last two oncologists did not remember the doses they usually prescribe for VTE high-risk patients. For the duration of prescribing prophylactic anticoagulants, six participants reported that they prescribed thromboprophylactic agents for six months (appropriate according to clinical guidelines), and four participants reported that they prescribed anticoagulants for three months. Other oncologists (n = 4) reported that they prescribe thromboprophylaxis during the whole period of chemotherapy treatment, and the last four participants reported that they decide the duration of thromboprophylaxis according to the patient's status (active disease, deterioration). In addition to thromboprophylaxis, two oncologists reported that they usually advise their patients to avoid prolonged immobilization.

"*I give high-risk cancer patients prophylactic anticoagulants. The drug that I prescribe in such cases is Clexan 50 mg per kg twice daily and continues till the end of chemotherapy because this period is the risky period for developing VTE. In addition, I advise these patients to avoid prolonged immobilization*" (P2)

"*Patients on hormonal therapy are at high risk for developing thrombosis, so I prescribe them aspirin 81mg per day for the duration of treatment as a prophylaxis*" (P4)

"*I consult a cardiologist for prescribing proper prophylaxis*" (P7)

"*I prescribe anticoagulants only when the patient develops a case of venous thromboembolism because of the side effects of treatment*" (P12)

## Advantages of VTE risk assessment among cancer patients

Three of the interviewed oncologists did not perceive any advantage in assessing VTE risk among cancer patients. Meanwhile, 28 of the participating oncologists reported one or more benefits of VTE risk assessment. The reported advantages of VTE risk assessment were mainly patient-related (n = 24) through reducing morbidity and mortality from VTE. Furthermore, eight participants considered VTE risk assessment to have hospital-related advantages by reducing hospital admission rates and the cost of treatment if VTE develops. The last six participants reported that VTE risk assessment process benefits oncologists because it saves their time, improves their relationship with their patients, and prevents the delay in starting chemotherapy that usually occurs when patients develop VTE.

"*I do not think that VTE risk assessment is a worthy process because prophylactic anticoagulants have many drug interactions*" (P13)

"*To prevent thromboembolic events and prevent complications of disease like pulmonary embolism, which is a fatal condition. Risk assessment for cancer patients will decrease the load on the hospital and the need for admission due to DVT*" (P8)

"*Cancer patients are tired, especially those with high risk like gastric and pancreatic cancer, so we must avoid VTE because VTE in such a situation will delay starting chemotherapy (It will interrupt my job, and cancer may progress), so risk assessment is a vital process. Also, when we conduct a risk assessment, we will decrease the load on our hospital*" (P28)

## Pharmacists' role in the prevention of VTE

Most participated oncologists (n = 24) reported that pharmacists have a significant role in preventing cancer-related thrombosis; three reported more than one role for pharmacists. The primary reported roles for pharmacists in the prevention of VTE include educating patients about the dispensed therapy (n = 3), helping oncologists in calculating VTE risk score (n = 3), and helping oncologists in prescribing a safe and effective anticoagulant therapy (n = 21). The last role can be achieved by calculating the prophylactic anticoagulant dose (n = 11), checking for drug-drug interactions (n = 9), providing oncologists with updated information about chemotherapeutic agents, especially those with thrombotic risks (n = 7), and adjusting the anticoagulant dose for patients with end-organ damage (n = 3).On the other hand, some oncologists reported that pharmacists either have no role (n = 2) or their role is limited to dispensing the prescribed medications to prevent VTE (n = 5).

"*Pharmacists must educate patients about side effects of cancer treatment, especially hormonal treatment, which may increase thrombotic events and cause VTE*" (P4)

"*I think the main role of pharmacists is to help physicians prescribe the best treatment with proper dose. They also can help in checking for drug interactions, especially for patients who use multiple medications*" (P7)

"*Pharmacists only provide and dispense treatment that we prescribe for the patients*" (P31)

"*Pharmacists can help us by calculating risk score for patients. They can also calculate the proper anticoagulant dose for the patient. They can provide us with updated information about drugs*" (P15)

## Discussion

The result of the present study showed that the majority of current study participants reported that VTE is commonly detected among patients with pancreatic and gastric cancers. This finding is in line with the current literature that considers patients with pancreatic and gastric cancer to be at the highest risk for developing VTE [19]. Meanwhile, nearly 1/4 of the interviewed oncologists reported that they commonly diagnose VTE among patients with breast and prostate cancer. In contrast to this finding, a meta-analysis of studies in many developed and developing countries found that breast and prostate cancers are the least likely cancers to increase the risk of VTE [20]. The difference between the current study results and that of the meta-analysis could be attributed to the poor adherence of most Iraqi oncologists to VTE prophylaxis guidelines as shown in the results of the current study and in other studies that conducted

in different Iraqi governorates (Al-Najaf and Al-Diwanyia) [21,22]. In addition, the current study findings (detection of VTE among patients with breast and prostate cancer) may raise attention to the possibility of a higher prevalence of VTE among Iraqi cancer patients than among those living in other countries. However, this assumption must be verified and explained by conducting well-designed clinical studies.

The results of the present study showed that the majority of participating oncologists agreed on the benefits of VTE risk assessment to the patient (reducing morbidity and mortality), to the oncologists (improving oncologists relationship with their patients), and even to the healthcare system (reducing costs of VTE-related hospital admission and treatment). All of the advantages mentioned above of thromboprophylaxis are highly expected since DVT is a life-threatening condition that poses a significant health and economic burden in hospitals [23]. Despite the high agreement of the interviewed oncologist on the benefits of VTE risk assessment, nearly two-thirds of them reported that they did not conduct any VTE risk assessment for their cancer patients. Similarly, Martin and colleagues found that 67% of oncologists in Chicago, USA, did not conduct VTE risk assessment risk for cancer patients [11]. Meanwhile, most of the current study participants considered VTE risk assessment time-consuming, making it difficult to perform during the daily workload on oncologists. This excuse may indicate the lack of oncologists' knowledge about the recommended method for VTE risk assessment, the Khorana score, which can be done easily and quickly [24–27]. The lack of oncologists' knowledge of VTE risk assessment is further confirmed by the fact that most of the participating oncologists who reported conducting VTE risk assessment did so by using a Wells score and/ or clinical examination. Both of the aforementioned measures can be used to confirm the diagnosis of VTE but not to assess the risk of developing VTE [28–30]. According to all of the above, it is easily concluded that the lack of oncologists' knowledge with guideline recommendations about thrombo-prophylaxis is the main reason behind neglecting VTE risk assessment. Therefore, it is highly recommended that Iraqi oncologists be provided with the latest guidelines about VTE risk assessment [24–26].

In addition, the results of the current study showed that only four (13%) of the participating oncologists reported the use of the recommended Khorana score. This finding was very close to a recent study conducted in the Middle Euphrates Cancer Center, Najaf Governorate, Iraq, in which only 4% of oncologists in that center use VTE prediction scores for their cancer patients [21]. This underutilization of validated scores for prediction of VTE among cancer patients raises a significant problem in the current clinical practice by most oncologists that must be addressed as soon as possible by policymakers in the Iraqi Ministry of Health (MOH). Meanwhile, such problem (underutilization of validated scores) is common in clinical practice not only in Iraq but also in the USA [31]. Therefore, increasing oncologists' awareness (through lectures and training workshops) about the importance of using the Khorana score for VTE risk assessment among ambulatory cancer patients is highly recommended. In addition, providing oncologists with Khorana score calculators can further enhance their clinical practice through adherence to VTE risk assessment guidelines by helping them to remember scores' parameters.

Despite the discouragement in adhering to VTE prophylaxis guidelines by most interviewed oncologists, more than half of them reported prescribing thromboprophylaxis for cancer patients with a high risk of developing VTE. However, anticoagulants were rarely prescribed in appropriate prophylactic doses and for appropriate periods. This action further confirms the limited competence of participating oncologists in thromboprophylaxis for cancer patients.

According to the study results, Rivaroxaban was the most commonly chosen drug for thromboprophylaxis among cancer patients. This favor toward Rivaroxaban is reasonable due

to its non-inferiority as compared to Enoxaparin in the prevention of VTE [32], its good safety and tolerability [33,34], besides its availability in an oral dosage form which renders it more accessible for patients [32].

Most oncologists who participated in the current study agreed on the significant role of pharmacists in preventing cancer-related thrombosis. They reported that pharmacists could prevent VTE by supporting oncologists in calculating VTE risk scores and also in prescribing a safe and effective anticoagulant therapy through checking for drug-drug interactions, adjusting the anticoagulant dose for patients with end-organ damage, and patient education about the dispensed thromboprophylaxis therapy. Similarly, Kandemir and colleagues found that clinical pharmacists are in a unique position to contribute to anticoagulant treatment by identifying and solving drug-related problems (such as the selection of appropriate drugs and doses) and assessing VTE risk among ambulatory cancer patients [35]. In addition, a recent study conducted in an Ambulatory cancer center in Ontario, Canada, found a significant benefit of incorporating VTE risk assessment into pharmacist practice by fostering inter-professional communication within the oncology care team, especially when initiating thromboprophylaxis in eligible patients [36].

Due to the current study's qualitative nature, its results may be slightly biased toward the bright side due to social desirability of oncologists during interviews [37]. To reduce such type of bias, a strategy of building rapport with participants was used before the interview. Anyhow, further observational studies are needed to accurately assess the daily clinical practice of oncologists in Iraqi oncology centers.

## Conclusion

Cancer-associated thrombosis is commonly diagnosed among Iraqi cancer patients. Most participating oncologists reported benefits of VTE risk assessment to the patient by reducing morbidity and mortality and to the healthcare institution by lowering costs of treating VTE and its complications. Meanwhile, assessing VTE risk for ambulatory cancer patients is rarely conducted by oncologists working at Oncology centers in Baghdad, Iraq. The prophylactic anticoagulants were rarely prescribed in appropriate dose and/or duration for patients at high risk of VTE. Increasing the awareness of oncologists through lectures and training workshops about the importance of using the Khorana score for VTE risk assessment among ambulatory cancer patients can enhance their clinical practice. Pharmacists can help oncologists follow thromboprophylaxis guidelines by calculating VTE risk score and recommending a safe and effective dose of appropriate prophylactic anticoagulant.

## Supporting information

**S1 Appendix. Interview guide.**
(DOCX)

**S1 Table. Characteristics of study participants.**
(DOCX)

## Author Contributions

**Conceptualization:** Ehab Mudher Mikhael.

**Formal analysis:** Marwa Akram Tariq.

**Investigation:** Marwa Akram Tariq.

**Methodology:** Ehab Mudher Mikhael.

**Project administration:** Marwa Akram Tariq.

**Resources:** Marwa Akram Tariq.

**Validation:** Ehab Mudher Mikhael.

**Writing – original draft:** Marwa Akram Tariq.

**Writing – review & editing:** Ehab Mudher Mikhael.

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
