## [Decision Letter · Decision Letter 0]

11 Jun 2024

PONE-D-24-15694Venous Thromboembolism Risk Assessment in Cancer Patients: A Qualitative StudyPLOS ONE

Dear Dr. Mikhael,

Thank you for submitting your manuscript to PLOS ONE. After careful consideration, we feel that it has merit but does not fully meet PLOS ONE’s publication criteria as it currently stands. Therefore, we invite you to submit a revised version of the manuscript that addresses the points raised during the review process.

We look forward to receiving your revised manuscript.

Kind regards,

Theresa Ukamaka Nwagha, M.B.B.S., M.P.H., FMCPath, M.Sc., M.D.,

Academic Editor

PLOS ONE

4. PLOS requires an ORCID iD for the corresponding author in Editorial Manager on papers submitted after December 6th, 2016. Please ensure that you have an ORCID iD and that it is validated in Editorial Manager. To do this, go to ‘Update my Information’ (in the upper left-hand corner of the main menu), and click on the Fetch/Validate link next to the ORCID field. This will take you to the ORCID site and allow you to create a new iD or authenticate a pre-existing iD in Editorial Manager. Please see the following video for instructions on linking an ORCID iD to your Editorial Manager account: https://www.youtube.com/watch?v=_xcclfuvtxQ.

5. Please amend your list of authors on the manuscript to ensure that each author is linked to an affiliation. Authors’ affiliations should reflect the institution where the work was done (if authors moved subsequently, you can also list the new affiliation stating “current affiliation:….” as necessary).

6. Please amend your manuscript to include your abstract after the title page.

8. We note that you have referenced (Tariq MA, Mikhael EM. Venous thromboembolism (VTE) risk assessment among newly diagnosed ambulatory cancer patients: A cross-sectional study. Unpublished work) which has currently not yet been accepted for publication. Please remove this from your References and amend this to state in the body of your manuscript: (ie “Bewick et al. [Unpublished]”) as detailed online in our guide for authors

Reviewers' comments:

Reviewer's Responses to Questions

**Comments to the Author**

1. Is the manuscript technically sound, and do the data support the conclusions?

Reviewer #1: Partly

Reviewer #2: Partly

2. Has the statistical analysis been performed appropriately and rigorously? 

Reviewer #1: No

Reviewer #2: I Don't Know

3. Have the authors made all data underlying the findings in their manuscript fully available?

Reviewer #1: Yes

Reviewer #2: Yes

4. Is the manuscript presented in an intelligible fashion and written in standard English?

Reviewer #1: Yes

Reviewer #2: Yes

5. Review Comments to the Author

Reviewer #1: The study addresses a relevant and underexplored area in oncology practice. The focus on an Iraqi context adds originality and contributes to the global understanding of VTE risk assessment practices. Ethical approval and consent processes are mentioned, but more detail on how confidentiality and participant rights were protected would strengthen the ethical rigor. The manuscript is generally well-written, but minor grammatical errors and redundancies should be addressed to enhance readability.

Suggested Revisions:

1. Provide details on written informed consent and measures to ensure ethical rigor.

2. Clarify the approach to ensuring intercoder reliability.

3. Address minor grammatical errors and improve overall readability.

4. Enhance the discussion of implications for clinical practice and policy.

Reviewer #2: This is a relevant study as it examines perceptions and attitude towards validated methodology for assessing risk in cancer patients. The findings are important for guiding interventions that will improve health care out comes in oncology patients. This is particularly important in settings where there may be a limited number of oncology specialists.

The title is misleading as it is a statement on the methodology or study design rather than the findings of the study.

The author should run a spell check to correct words which have no space between them.

The data and questions in the data tool (Section B) are more related to practice rather than perception. Questions such as How often? How do you assess? What is your action? are not well elaborated. There is much quantitative data and it is not clear how this was obtained. The study appears to have both quantitative and qualitative data. It will be good for the author to elaborate which results focus on practice and which data are for perception, how this was collected and analyzed.

In the results section on Dealing with Cancer patients with high risk of VTE: how did the author assess appropriateness of dosage? Was this part of the objective of the study?

The conclusions are not supported by data from this study. The author concludes that limited competence as the main reason for poor compliance to guidelines for guidelines. How was compliance measured, and how did the author determine that there was poor compliance to thromboprophylaxis guidelines? Again, this as stated in the objective, was a study on perception and practice towards risk assessment for VTE.

6. PLOS authors have the option to publish the peer review history of their article (what does this mean?). If published, this will include your full peer review and any attached files.

Reviewer #1: No

Reviewer #2: No

---

## [Author Response · Author response to Decision Letter 0]

17 Jun 2024

Dear reviewer 1, 

First of all, we want to express our gratitude for the time and effort you put into providing insightful feedback on our manuscript.

It is our pleasure to inform you that all of your suggestions and comments were carefully reviewed. We did our best to make nearly all of the suggested revisions in our manuscript. We believe that these changes have strengthened the overall quality and clarity of the manuscript, and we are confident that it now meets the standards of PLOS One. 

We look forward to hear your feedback on the revised version and hope that it meets your expectations.

Thank you once again for your valuable comments and suggestions.

Answers to reviewer 1 Comments

1. In abstract (results): However, this section can be improve in its quantitative data presentation to be more consistent; for example, replacing "64% of participating oncologists emphasized that they didn't conduct VTE risk assessments" with "64% of oncologists reported not conducting VTE risk assessments."

Response: A revision was done based on your valuable comment. Please refer to the highlights in the following statement.

64% of participating oncologists reported that they did not conduct VTE risk assessments for their cancer patients.

2. Abstract: the conclusion could also add implications for practice and future research, such as the need for enhanced training and support for oncologists in VTE risk assessment and management.

Response: A revision was done based on the given comment. Please refer to the highlighted statement. 

Educating and training oncologists about VTE risk assessment is recommended to enhance their practice in thromboprophlaxis.

3. Introduction: The citations support the statements made, and the gap in knowledge regarding clinicians' adherence to guidelines is identified, however it may be relevant to identify and state who are considered as "high risk patients".

Response: A new statement was added to identify who are considered as high risk patients. Please refer to the highlighted statement. 

Therefore, current guidelines recommend VTE prevention for high-risk patients (who have about 7% risk to develop VTE) after assessing VTE and the bleeding risk of cancer patients [7,8].

4. Results: Authors should consider that it may be more informative to have tables to summarize the results of the different cancers that were found to have VTE under the heading "VTE among Cancer patients", as well as the drugs administrated

Response: New tables (table 4 and table 5) were added. Please refer to the highlighted tables in the attached file. 

5. The discussion could be improved from a deeper analysis of the implications of these findings for clinical practice and policy.

Response: More details were added to discussion. Please refer to the following highlighted statements.

This underutilization of validated scores for prediction of VTE among cancer patients raises a significant problem in the current clinical practice by most oncologists that must be addressed as soon as possible by policymakers in the Iraqi Ministry of Health (MOH). Meanwhile, such problem (underutilization of validated scores) is common in clinical practice not only in Iraq but also in the USA [31]. Therefore, increasing oncologists' awareness (through lectures and training workshops) about the importance of using the Khorana score for VTE risk assessment among ambulatory cancer patients is highly recommended. In addition, providing oncologists with Khorana score calculators can further enhance their clinical practice through adherence to VTE risk assessment guidelines by helping them to remember scores' parameters.

6. The potential biases in the qualitative study, such as social desirability bias, are acknowledged, but strategies to mitigate these biases during data collection could be elaborated.

Response: More details were added. Please refer to the following highlighted statement.

To reduce such type of bias, a strategy of building rapport with participants was used before the interview.

7. Ethical approval and consent processes are mentioned, but more detail on how confidentiality and participant rights were protected would strengthen the ethical rigor.

Response: Details were added. Please refer to the following highlighted statement

To achieve confidentiality, all participants were interviewed in a quiet area either in their office or in the consultation area while keeping a distance of at least 5 meters from other physicians or patients.

8. Provide details on written informed consent and measures to ensure ethical rigor.

Response: Indeed, we obtained verbal informed consent from study participants, and this was already mentioned in our manuscript. However, we added more details about the issue of verbal informed consent. Please refer to the following highlighted statement.

The current study was ethically approved by the ethical committee at the College of Pharmacy/University of Baghdad (approval number of RECAUBCP10620238). The ethical committee accepted waiving written informed consent due to cultural issues and potential fear from signing documents by most Iraqi individuals. Thus, it was requested that the authors obtain verbal informed consent from study participants.

9. Clarify the approach to ensuring intercoder reliability.

Response: Details were added. Please refer to the following highlighted statements.

The coding procedure was established by the first study author (MSc candidate in clinical pharmacy) and reviewed by the second study author (PhD in clinical pharmacy). Any discrepancy in coding was solved after negotiation and reaching consensus. The coding procedure was started by a thorough reading of each interview transcript. A codebook was developed to ensure consistent coding across interviews.

10. Address minor grammatical errors and improve overall readability.

Response: The manuscript was checked manually for grammatical errors. All found errors were corrected. Besides that we corrected all words which have no space between them. Then, the final manuscript version was further checked by subscribed grammarly application and all suggestions of the application were made when feasible. 

11. Enhance the discussion of implications for clinical practice and policy.

Response: Details about this subject were added. Please refer to the following highlighted statements.

This underutilization of validated scores for prediction of VTE among cancer patients raises a significant problem in the current clinical practice by most oncologists that must be addressed as soon as possible by policymakers in the Iraqi Ministry of Health (MOH). Meanwhile, such problem (underutilization of validated scores) is common in clinical practice not only in Iraq but also in the USA [31]. Therefore, increasing oncologists' awareness (through lectures and training workshops) about the importance of using the Khorana score for VTE risk assessment among ambulatory cancer patients is highly recommended. In addition, providing oncologists with Khorana score calculators can further enhance their clinical practice through adherence to VTE risk assessment guidelines by helping them to remember scores' parameters.

Dear reviewer 2, 

First of all, we want to express our gratitude for the time and effort you put into providing insightful feedback on our manuscript.

It is our pleasure to inform you that all of your suggestions and comments were carefully reviewed. We did our best to make nearly all of the suggested revisions in our manuscript. We believe that these changes have strengthened the overall quality and clarity of the manuscript, and we are confident that it now meets the standards of PLOS One. 

We look forward to hear your feedback on the revised version and hope that it meets your expectations.

Thank you once again for your valuable comments and suggestions.

Answers to reviewer 2 Comments

1. The title is misleading as it is a statement on the methodology or study design rather than the findings of the study.

Response: The title was revised to be more comprehensive 

Determining the Perceptions and Practices of Oncologists Regarding Venous Thromboembolism Risk Assessment in Ambulatory Cancer Patients: A Qualitative Study

2. The author should run a spell check to correct words which have no space between them.

Response: Done

3. The data and questions in the data tool (Section B) are more related to practice rather than perception. There is much quantitative data and it is not clear how this was obtained. The study appears to have both quantitative and qualitative data. It will be good for the author to elaborate which results focus on practice and which data are for perception, how this was collected and analyzed.

Response: Yes, you are right. Our questions are more related to practice. Details about which questions focus on practice and which for perception were added to method section. Details about the method for obtaining quantitative results were also added to methods. Please refer to the following highlighted statements,

The validated interview guide consisted of eight questions, five (question 1 to 5) designed to assess oncologists practice in detecting and preventing VTE. All obtained answers regarding the type, dose, and duration of prophylactic anticoagulant were evaluated for appropriateness according to the recommendations in thromboprophylaxis guidelines [7,8]. Two questions were developed to assess oncologists' perceptions of the advantages of thromboprophylaxis and pharmacists' role in thromboprophylaxis. The last question was designed to capture any additional comments from participants.

In addition, most obtained numbers in result section for example those who reported conducting VTE risk assessment were calculated manually based on responses from study participants. Adding numbers in qualitative studies has many advantages

1. It contributes to the internal generalizability of qualitative researchers’ claims

2. Quantitative data can help in identifying patterns that are not apparent simply from the unquantitized qualitative data

3. Quantitative data help to adequately present evidence for interpretations and to counter claims that researchers have simply cherry picked the data for instances that support these interpretations

Reference: Maxwell JA. Using Numbers in Qualitative Research. Qualitative Inquiry. 2010; 16(6) 475–482.DOI: 10.1177/1077800410364740

4. In the results section on Dealing with Cancer patients with high risk of VTE: how did the author assess appropriateness of dosage? Was this part of the objective of the study?

Response: Details were added to method section (please refer to the above). This part is indirectly present in study objectives that is related to exploring oncologists practice in VTE prevention (aim of study was slightly revised). Please refer to the highlighted change

Therefore, the current study aimed to explore in depth the practices and perceptions of oncologists working in oncology centers in Baghdad, Iraq, regarding VTE, its risk assessment, and prevention among ambulatory cancer patients.

5. The conclusions are not supported by data from this study. The author concludes that limited competence as the main reason for poor compliance to guidelines for guidelines. How was compliance measured, and how did the author determine that there was poor compliance to thromboprophylaxis guidelines? Again, this as stated in the objective, was a study on perception and practice towards risk assessment for VTE. 

Response: Conclusion was revised to be more compatible with study results. Please refer to the highlighted statements.

Cancer-associated thrombosis is somewhat prevalent among Iraqi cancer patients. Most participating oncologists reported benefits of VTE risk assessment to the patient by reducing morbidity and mortality and to the healthcare institution by lowering costs of treating VTE and its complications. Meanwhile, assessing VTE risk for ambulatory cancer patients is rarely conducted by oncologists working at Oncology centers in Baghdad, Iraq. The prophylactic anticoagulants were rarely prescribed in appropriate dose and/or duration for patients at high risk of VTE. Increasing the awareness of oncologists through lectures and training workshops about the importance of using the Khorana score for VTE risk assessment among ambulatory cancer patients can enhance their clinical practice. Pharmacists can help oncologists follow thromboprophylaxis guidelines by calculating VTE risk score and recommending a safe and effective dose of appropriate prophylactic anticoagulant.

---

## [Decision Letter · Decision Letter 1]

30 Oct 2024

PONE-D-24-15694R1Determining the Perceptions and Practices of Oncologists Regarding Venous Thromboembolism Risk Assessment in Ambulatory Cancer Patients: A Qualitative StudyPLOS ONE

Dear Dr. Mikhael,

Thank you for submitting your manuscript to PLOS ONE. After careful consideration, we feel that it has merit but does not fully meet PLOS ONE’s publication criteria as it currently stands. Therefore, we invite you to submit a revised version of the manuscript that addresses the points raised during the review process.

We look forward to receiving your revised manuscript.

Kind regards,

Maher Abdelraheim Titi

Academic Editor

PLOS ONE

Reviewers' comments:

Reviewer's Responses to Questions

**Comments to the Author**

1. If the authors have adequately addressed your comments raised in a previous round of review and you feel that this manuscript is now acceptable for publication, you may indicate that here to bypass the “Comments to the Author” section, enter your conflict of interest statement in the “Confidential to Editor” section, and submit your "Accept" recommendation.

Reviewer #2: (No Response)

Reviewer #3: (No Response)

2. Is the manuscript technically sound, and do the data support the conclusions?

Reviewer #2: Partly

Reviewer #3: Partly

3. Has the statistical analysis been performed appropriately and rigorously? 

Reviewer #2: Yes

Reviewer #3: I Don't Know

4. Have the authors made all data underlying the findings in their manuscript fully available?

Reviewer #2: Yes

Reviewer #3: Yes

5. Is the manuscript presented in an intelligible fashion and written in standard English?

Reviewer #2: Yes

Reviewer #3: Yes

6. Review Comments to the Author

Reviewer #2: The data presented in Table 4 is the number of oncologists who identified or selected that particular malignancy as being associated with VTE. The data does not represent the frequency of actual occurrence of VTE in patients with malignancy. The author may want to clarify this in the results.

Reviewer #3: This is a study that aims to understand the perceptions and practices of oncologist regarding risk of VTE in ambulatory cancer patients as well as primary prophylaxis prescription patterns in Baghdad, Iraq.

The question guide was developed and reviewed by a panel of experts with interviews conducted 11/2023 – 1/2024.

Sample size = 31 oncologists

Overall, 64% of the interviewed oncologists do not assess VTE risks; however, 71% have diagnosed VTE in their cancer patients. Of the 58% that have prescribed primary thromboprophylaxis, dosing was inappropriate for most prescriptions.

I have the following concerns related to the study.

1. The conclusion that CAT is prevalent in Iraqi cancer patients is inaccurate based on the data presented. The authors note that 71% of oncologists diagnose VTE in their patients. This suggests that most oncologists in the study have diagnosed VTE in cancer. Not that VTE is prevalent in cancer. To assess prevalence, you would need a denominator of all patients cared for by the 31 oncologists and calculate prevalence. Please revise.

2. The statement that most prophylaxis prescriptions were inappropriate could use data for support in the abstract, especially given this is a conclusion of the study.

3. The introduction Paragraph 1-2 has jumping concepts. Recommended Modification for flow and clarity:

Venous thromboembolism (VTE) is a serious, potentially, life-threatening condition and a major cause of mortality and morbidity [1][6]. Approximately 20% of all newly diagnosed cases of VTE are cancer patients [2] and an estimated 8% of cancer patients develop VTE within one year after diagnosis or progression of their malignancy [4,5]. Therefore, current guidelines recommend primary prophylaxis for VTE prevention in high-risk patients after assessing the competing risk of bleeding [7,8]. The prediction of VTE in cancer patients often involves the use of risk assessment models or scores (e.g., Khorana Score) based on the presence of certain clinical and laboratory factors [9].

Despite available evidence and existing guideline recommendations for VTE risk assessment of ambulatory cancer patients with primary thromboprophylaxis for identified high-risk patients, several studies found limited utilization of both risk assessment and prescription of primary prophylaxis in clinical practice [10,11,12]. However, several questions remain regarding assessment of risk and prescription of prophylaxis. First, understanding clinician knowledge and understanding regarding guideline recommendations. Second, frequency of VTE risk assessment and prescription of primary thromboprophylaxis for identified high-risk patients [10]. Therefore, the current study aimed to explore the practices and perceptions of oncologists working in Baghdad, Iraq, regarding cancer-associated VTE, its risk assessment, and prevention among ambulatory cancer patients.

4. Methods – change “validated by a panel of experts” to reviewed or endorsed by a panel of experts

5. Table 2 can be moved to the supplement

6. For the result section “VTE among Cancer Patients”

a. Did the oncologists participating subspecialize? For example, were there oncologist that focus only on specific types of cancer? This will bias their answers.

b. For example, VTE is less common in patients with prostate cancer, but will be more common in pancreatic cancer etc. If you have a GU oncologist, they may be seeing VTE infrequently.

7. Decrease use of the phrase “on the other hand”

8. For the abstract and results section stating that most patients are prescribed inappropriate dosing of prophylaxis, I recommend changing the result/interpretation to “of oncologists that prescribe primary thromboprophylaxis, most prescribe inappropriate doses compared to those recommended by guidelines or prior randomized controlled trials” (cite guidelines, AVERT, CASSINI).

9. First line of the discussion is not supported by the data. The results stated that: 14 oncologists state that sometimes (not many) their patients develop and 8 stated that their patients commonly develop VTE. Sometimes does not = many. Also, this is not the main purpose of this study. This study is not to determine if VTE is common in cancer. It is to determine knowledge of VTE risk assessment and prophylaxis prescribing patterns, unless I misunderstood. Recommend deleting this sentence/conclusion throughout the paper.

10. Regarding type of cancer associated with VTE, again, this could be baised by the practice composition of the oncologist. I would not make this a key focus of the study. I recommend deleting all of paragraph 1 of the discussion. It is not supported by the study.

7. PLOS authors have the option to publish the peer review history of their article (what does this mean?). If published, this will include your full peer review and any attached files.

Reviewer #2: No

Reviewer #3: No

---

## [Author Response · Author response to Decision Letter 1]

6 Nov 2024

Answers to reviewer 2 Comment

The data presented in Table 4 is the number of oncologists who identified or selected that particular malignancy as being associated with VTE. The data does not represent the frequency of actual occurrence of VTE in patients with malignancy. The author may want to clarify this in the results.

Response: A revision was done based on your valuable comment. We revise this to explicitly state that the data represent the number of oncologists who identified each type of cancer as being associated with VTE risk and not the frequency of VTE occurrences in these patient populations.

''In addition, the study participants reported that patients with certain types of cancer have a high risk of developing VTE (Table 4) such as gastric (n=17), pancreatic (n=14), breast (n=7), lung (n=7), colorectal (n=6), gynecological (n=4), pelvic (n=4), prostate (n=2), liver (n=2), kidney (n=1), bladder (n=1), lymphoma (n=1), and brain cancer (n=1). It is important to note that these findings reflect the observations of the participating oncologists among their patients in daily practice, rather than the actual incidence of VTE in these populations.''

Answers to reviewer 3 Comments

1. The conclusion that CAT is prevalent in Iraqi cancer patients is inaccurate based on the data presented. The authors note that 71% of oncologists diagnose VTE in their patients. This suggests that most oncologists in the study have diagnosed VTE in cancer. Not that VTE is prevalent in cancer. To assess prevalence, you would need a denominator of all patients cared for by the 31 oncologists and calculate prevalence. Please revise.

Response: A revision was done based on your valuable comment.

''Conclusion: Cancer-associated thrombosis (CAT) is commonly diagnosed among Iraqi cancer patients. VTE risk assessment for ambulatory cancer patients is rarely conducted by oncologists working at Oncology centers in Baghdad, Iraq.''

2. The statement that most prophylaxis prescriptions were inappropriate could use data for support in the abstract, especially given this is a conclusion of the study.

Response: A new statement was added to abstract. 

''58% of oncologists reported that they prescribe thromboprophylaxis for high-risk patients; meanwhile, only 11% of them reported prescribing anticoagulants in a doses similar to that reported by thromboprophylaxis guidelines.''

3. The introduction Paragraph 1-2 has jumping concepts. Recommended Modification for flow and clarity.

Response: The introduction was revised according to your valuable comment.

''Venous thromboembolism (VTE) is a serious, potentially life-threatening condition and a major cause of mortality and morbidity [1][2]. Approximately 20% of all newly diagnosed cases of VTE are cancer patients [3], and an estimated 8% of cancer patients develop VTE within one year after diagnosis or progression of their malignancy [4,5]. Therefore, current guidelines recommend primary prophylaxis for VTE prevention in high-risk patients after assessing the competing risk of bleeding [6,7]. The prediction of VTE in cancer patients often involves the use of risk assessment models or scores (e.g., Khorana Score) based on the presence of certain clinical and laboratory factors [8]. 

Despite available evidence and existing guideline recommendations for VTE risk assessment of ambulatory cancer patients with primary thromboprophylaxis for identified high-risk patients, several studies found limited utilization of both risk assessment and prescription of primary prophylaxis in clinical practice [9,10,11]. However, several questions remain regarding assessing risk and prescribing prophylaxis. First, understanding clinician knowledge and understanding regarding guideline recommendations. Second, the frequency of VTE risk assessment and prescription of primary thromboprophylaxis for identified high-risk patients [10]. Therefore, the current study aimed to explore the practices and perceptions of oncologists working in Baghdad, Iraq, regarding cancer-associated VTE, its risk assessment, and prevention among ambulatory cancer patients.''

4. Methods – change “validated by a panel of experts” to reviewed or endorsed by a panel of experts.

Response: the statement was changed according to your valuable suggestion.

''The guide was reviewed by a panel of experts (Three academic clinical pharmacists with experience in qualitative research and one oncologist).''

5. Table 2 can be moved to the supplement

Response: Table 2 were moved to Supporting information (supplement 2) (S2)

6. For the result section “VTE among Cancer Patients

 Did the oncologists participating subspecialize? For example, were there oncologist that focus only on specific types of cancer? This will bias their answers.

b. For example, VTE is less common in patients with prostate cancer, but will be more common in pancreatic cancer etc. If you have a GU oncologist, they may be seeing VTE infrequently.

Response: All participating oncologists hold a general medical or clinical oncology certificate and do not subspecialize in specific types of cancer. This means they treat a diverse range of cancer patients, which helps to provide a balanced perspective on the association between different cancers and the risk of VTE.

7. Decrease use of the phrase “on the other hand”

Response: We have carefully reviewed the manuscript and made adjustments to minimize its use

8. For the abstract and results section stating that most patients are prescribed inappropriate dosing of prophylaxis, I recommend changing the result/interpretation to “of oncologists that prescribe primary thromboprophylaxis, most prescribe inappropriate doses compared to those recommended by guidelines or prior randomized controlled trials” (cite guidelines, AVERT, CASSINI).

Response: It was revised. Please refer to the highlighted statement. 

''Of oncologists that prescribe primary thromboprophylaxis, most oncologists (n=10) prescribe inappropriate doses compared to those recommended by guidelines or prior randomized controlled trials [17,18]''

9. First line of the discussion is not supported by the data. The results stated that: 14 oncologists state that sometimes (not many) their patients develop and 8 stated that their patients commonly develop VTE. Sometimes does not = many. Also, this is not the main purpose of this study. This study is not to determine if VTE is common in cancer. It is to determine knowledge of VTE risk assessment and prophylaxis prescribing patterns, unless I misunderstood. Recommend deleting this sentence/conclusion throughout the paper.

Response: the statement in the discussion was deleted according to your valuable comment. 

''The result of the present study showed that most of the participating oncologists reported that many of their cancer patients develop VTE at some point in their life. Similar to the current finding, a large cohort study found that the incidence of VTE is high among cancer patients in California, USA [18].''

10. Regarding type of cancer associated with VTE, again, this could be baised by the practice composition of the oncologist. I would not make this a key focus of the study. I recommend deleting all of paragraph 1 of the discussion. It is not supported by the study.

Response: This information was obtained from physicians, and as we mentioned previously, we do not have a subspecialist in oncology. Therefore, we believe the data we collected is free from bias. For this reason, we chose to retain this information rather than remove it.

Additionally, we identified an unusual finding that could potentially serve as a foundation for future research, which is why we decided to keep this section. In response to the reviewer’s suggestion, however, we have removed the first part of the initial paragraph.

Furthermore, the current study findings (detection of VTE among patients with breast and prostate cancer) may raise attention to the possibility of a higher prevalence of VTE among Iraqi cancer patients than among those living in other countries. However, this assumption must be verified and explained by conducting well-designed clinical studies.

---

## [Decision Letter · Decision Letter 2]

18 Dec 2024

Determining the Perceptions and Practices of Oncologists Regarding Venous Thromboembolism Risk Assessment in Ambulatory Cancer Patients: A Qualitative Study

PONE-D-24-15694R2

Dear Dr. Mikhael,

We’re pleased to inform you that your manuscript has been judged scientifically suitable for publication and will be formally accepted for publication once it meets all outstanding technical requirements.

Kind regards,

Maher Abdelraheim Titi

Academic Editor

PLOS ONE

Additional Editor Comments (optional):

Reviewers' comments:

Reviewer's Responses to Questions

**Comments to the Author**

1. If the authors have adequately addressed your comments raised in a previous round of review and you feel that this manuscript is now acceptable for publication, you may indicate that here to bypass the “Comments to the Author” section, enter your conflict of interest statement in the “Confidential to Editor” section, and submit your "Accept" recommendation.

Reviewer #3: All comments have been addressed

2. Is the manuscript technically sound, and do the data support the conclusions?

Reviewer #3: Yes

3. Has the statistical analysis been performed appropriately and rigorously? 

Reviewer #3: Yes

4. Have the authors made all data underlying the findings in their manuscript fully available?

Reviewer #3: Yes

5. Is the manuscript presented in an intelligible fashion and written in standard English?

Reviewer #3: Yes

6. Review Comments to the Author

Reviewer #3: Thank you for addressing my comments. I have no further concerns at this time. I recommend publication.

7. PLOS authors have the option to publish the peer review history of their article (what does this mean?). If published, this will include your full peer review and any attached files.

Reviewer #3: No

---

## [Editor Report · Acceptance letter]

26 Dec 2024

PONE-D-24-15694R2 

PLOS ONE

Dear Dr. Mikhael, 

I'm pleased to inform you that your manuscript has been deemed suitable for publication in PLOS ONE. Congratulations! Your manuscript is now being handed over to our production team.

Kind regards, 

on behalf of

Dr. Maher Abdelraheim Titi 

Academic Editor

PLOS ONE
